# PRIME: PROTECT YOUR VIDEOS FROM MALICIOUS EDITING

## ABSTRACT

Over the years, video generation has experienced significant advancement. A variety of open-source models emerge, making it surprisingly easy to manipulate and edit videos with just a few simple prompts. While these cutting-edge technologies have gained huge popularity, they have also given rise to concerns regarding the privacy and portrait rights of individuals: malicious users can exploit these tools for deceptive or illegal purposes. Existing works on protecting images against generative models cannot be directly grafted to video protection, due to their efficiency and effectiveness limitations. Motivated by this, we introduce PRIME, a new methodology dedicated to the protection of videos from unauthorized editing via generative models. Our key idea is to craft highly transferable and robust perturbations, which can be efficiently added to the protected videos to disrupt their editing feasibility. We perform comprehensive evaluations using both objective metrics and human studies. The results indicate that PRIME only needs 8.3% GPU hours of existing state-of-the-art methods while achieving better protection results.

## 1 INTRODUCTION

Benefiting from the rapid development of generative AI technology, *video editing* has become remarkably simple and popular. Given an original video clip, users can freely add new elements or alter its style by just providing a few language prompts. Researchers have proposed a number of methods for efficient video editing (Khachatryan et al., 2023; Khandelwal, 2023; Qi et al., 2023). These solutions treat the video as a sequence of images arranged along the temporal dimension and leverage the state-of-the-art image editing techniques based on diffusion models (Ho et al., 2020; Song et al., 2021a;b) to modify each frame. They further need to ensure the coherence and consistency of these stacked images, employing techniques such as global attention constraints (Geyer et al., 2024) and latent feature constraints (Yang et al., 2023; Khachatryan et al., 2023). With these techniques, there are now a lot of video editing applications and online services (e.g., Runway Gen-1 (Run), Pika (Pik)), significantly enhancing our life quality and working efficiency.

Unfortunately, advanced video editing methods also open the door to potential misuse. Malicious users could download pristine videos from the Internet and then leverage video editing models to generate illegal or harmful content. For example, they can remove the clothes from the subjects in the videos to cause sexual innuendo. The international anti-human trafficking organization, Thorn (th), cooperated with AI companies (e.g., AWS AI, Civitai, and Hugging Face) to propose a white paper book (Tho) on protecting child sexual abuse against generative AI. Malicious users can also swap the faces of people in the videos to create and propagate fake content. The Princess of Wales's video is considered as AI generated (pri), which tarnishes the reputation of the British royal family.

It is critical to protect sensitive videos from malicious editing with existing generative models. A lot of research efforts have been dedicated to protecting images from unauthorized usage (Salman et al., 2023; Shan et al., 2023; Le et al., 2023; Rhodes et al., 2023; Zheng et al., 2023). They mainly introduce subtle and imperceptible perturbations on the static images, which can effectively compel the generative models to produce poor editing outcomes. In contrast, very rare studies are conducted for video protection. Since a video is composed of a sequence of images, a straightforward strategy is to apply the image protection methods to each frame of the target video. Unfortunately, this brings new issues in practice. (1) *Efficiency*. Existing image protection methods are quite time-consuming. This overhead is exacerbated when we try to craft perturbations for each video frame: even protecting

a short video clip of 2 seconds (30 fps) may take hours. This significantly restricts the practicality of existing solutions. (2) *Robustness*. During the video coding process, the codec is adopted to assign different compression ratios to individual frames to achieve the best trade-off between the video quality and size. This dynamic compression scheme can diminish the effect of the perturbation on each frame, leading to ineffective protections.

The gaps mentioned above motivate us to explore and design new protection methods dedicated to video protection. To this end, we introduce PRIME, a novel methodology to PRotect vIdeos from Malicious Editing. Our solution adds perturbations to each frame of the protected video to prevent the constraints, such as global attention, used in the video editing pipeline from rectifying the wrong features of the perturbed frames with the clean frames. We introduce the following innovative techniques to make the protection more efficient and robust. (1) The malicious users may use diverse editing methods with different pre-trained models, pipelines, and prompts, which are all agnostic to the defenders. Hence, the crafted perturbations must have high transferability (Papernot et al., 2016) across various models and editing methods. To improve the transferability among different LDMs and achieve the zero-shot ability, we consider the latent features during the diffusion process and the final outputs. (2) Perturbing each frame in the given video can cost a lot of time and GPU resources. We introduce two mechanisms to enhance the efficiency of the perturbation generation process: *fast convergence searching* helps us find a better image for each frame as the editing target; *early stage stopping* monitors the perturbation generating process and finds the optimal step to stop, which can yield the high-quality perturbation within a short time. (3) The compression algorithm used in the video codec is generally complex and lossy, making it impossible to maintain lossless perturbation after compression. Following an in-depth examination of existing video codec methods, we propose a new *anti dynamic compression* scheme to discretize perturbations within the pixel space, which can enhance the robustness of perturbations during compression.

Following the previous works of video editing (Geyer et al., 2024; Yang et al., 2023; Wu et al., 2023), we collect video clips from the Internet to evaluate the performance of our proposed solution. We consider two types of malicious editing scenarios: malicious Not-Safe-For-Work (NSFW) editing and malicious swapping editing. Due to the lack of ground-truth references, it is difficult to comprehensively and objectively assess the edited videos. To address this, we follow the evaluation methods in previous works (Geyer et al., 2024; Yang et al., 2023; Wu et al., 2023) to conduct a user study, wherein we engage volunteers to participate in surveys gauging the quality and preferences of the video. Based on the human evaluation, we show that existing video editing methods can produce high-quality videos for these two malicious purposes, and obtain 2.99 out of 5 and 3.17 out of 5 for the video quality scores, respectively. On the other hand, PRIME can significantly reduce the quality of the generated malicious videos (1.54 out of 5 and 2.44 out of 5 for two tasks, respectively), making them totally unusable. Overall, our contributions can be summarized as follows:

- We propose PRIME, a new black-box video protection method against malicious editing. We improve its transferability by simultaneously considering latent codings and generated images.

- PRIME is timing-efficient with our proposed *fast convergence searching* and *early stage stopping* mechanisms. It only costs about 8.3% GPU hours of the state-of-the-art solution Photoguard Salman et al. (2023) under the same setting.

- PRIME can combat dynamic compression from the lossy video codec with our proposed *anti dynamic compression* method. It increases about 8% of the bitrate for the protected videos compared with Photoguard.

- We perform a user study to show that PRIME demonstrates better protection performance and transferability than Photoguard for two malicious editing tasks.

## 2 RELATED WORKS

### 2.1 VIDEO EDITING WITH LDMS

A growing number of works (Parmar et al., 2023; Wu & Torre, 2023; Lin et al., 2024) focus on turning a latent diffusion model (LDM) (Rombach et al., 2022) into a zero-shot image editor. Such progress inspires the development of *video editing*, a subtask in the video generation domain (Blattmann et al., 2023; Chen et al., 2023; Ge et al., 2023). Different from the conventional video generation task, which usually only requires a conditional prompt to guide the generation process, video editing needs a source video and a guidance prompt as conditions. The edited video can be seen as a series of

images stacked along the time dimension. However, directly editing each frame of the given video will probably lead to different backgrounds or different poses for the foreground objects, due to the lack of pixel-level constrain. How to keep the frames of the edited video consistent is still an open problem. To keep the consistency between frames, cross-frame global attention (Geyer et al., 2024; Yang et al., 2023; Wu et al., 2023) is widely used in existing video editing frameworks based on various pre-trained LDMs. On the other hand, some frameworks (Yang et al., 2023; Khachatryan et al., 2023) adopt other conditions (e.g., depth, pose, edge) to better enhance consistency.

## 2.2 Protecting Images from Malicious Editing

It is a common practice for individuals to release and share images on public platforms. However, these images could be misused for illegal purposes. Researchers have proposed new methods to protect the images from two perspectives. First, attackers could edit these images to create harmful or fake content. The advance of LDMs can significantly facilitate this image manipulation process. The state-of-the-art defense against this threat is Photoguard (Salman et al., 2023) is the state-of-the-art solution. This method incorporates adversarial perturbations into images, effectively perplexing LDMs and preventing unauthorized editing. Second, attackers could collect the images from the Internet and use them without any authorization to fine-tune LDMs with some personalized techniques like DreamBooth (Ruiz et al., 2023). To prevent the intellectual property violations of these images, some works (Shan et al., 2023; Le et al., 2023; Rhodes et al., 2023; Zheng et al., 2023) add perturbations into images before releasing them to the public. With such perturbed images, the fine-tuned LDMs are only capable of producing low-quality results.

In this paper, we mainly focus on the first threat scenario, but in the context of video editing with LDMs. As currently there are no solutions dedicated to video protection, we mainly treat Photoguard as a baseline method. As discussed in Section 1, this method has severe efficiency and robustness drawbacks when applied to the video protection. This motivates us to design a new solution specifically against the video editing threat.

## 3 Preliminary

### 3.1 Latent Diffusion Model

We briefly describe the mechanism of LDMs and how they can be used to edit images. Videos can be edited in a similar way. A LDM contains an image encoder $\mathcal{E}$, a U-Net $\mathcal{U}$, and an image decoder $\mathcal{D}$. The image encoder projects a given image $x$ to its latent code $f_{\mathcal{E}} = \mathcal{E}(x)$. The image decoder projects a latent code to an image $x = \mathcal{D}(\mathcal{E}(x))$. The U-Net $\mathcal{U}$ accepts the latent codes $f_{\mathcal{E}}$ and is related to a diffusion process, which contains a noise-adding forward process and a denoising sampling process. For the forward process, given a time series $t = [1, \ldots, T]$, we have the following relation between clean $f_{\mathcal{E},0}$ and noisy $f_{\mathcal{E},t}^d$:

$$q(f_{\mathcal{E},t}^d | f_{\mathcal{E},0}) = \mathcal{N}(f_{\mathcal{E},t}^d; \sqrt{\bar{a}_t} f_{\mathcal{E},0}, (1 - \bar{a}_t)\mathbf{I}), t = [1, \ldots, T],$$

where $\mathcal{N}$ stands for a Gaussian distribution, $\bar{a}_t$ is a hyperparameter related to the diffusion process, $f_{\mathcal{E},0} = \mathcal{E}(x)$ is the clean latent, and $f_{\mathcal{E},t}^d$ is the noisy latent at time step $t$. For the sampling process, $f_{\mathcal{E},t-1}^s$ can be predicted with $\mathcal{U}(f_{\mathcal{E},t}^s, t, c_p)$ under DDIM sampling (Song et al., 2021a), where $c_p$ is a given condition based on a prompt and $f_{\mathcal{E},T}^s = f_{\mathcal{E},T}^d$ which is the boundary condition.

To edit an image, e.g., swapping the face of Joe Biden in an image $x$ with that of Donald Trump, we first obtain the latent code $f_{\mathcal{E},0} = \mathcal{E}(x)$. Then we add noise to $f_{\mathcal{E},0}$ and obtain $f_{\mathcal{E},T_1}^d$, where $T_1 \leq T$. With the prompt $c_p$: "*a photo of Donald Trump*", we sample $f_{\mathcal{E},t-1}^s$ from $\mathcal{U}(f_{\mathcal{E},t}^s, t, c_p)$ and $f_{\mathcal{E},T_1}^s = f_{\mathcal{E},T_1}^d$ step by step and get $f_{\mathcal{E},0}^s$, which is the clean latent for prompt $c_p$. Finally, we use $\mathcal{D}$ to convert $f_{\mathcal{E},0}^s$ to an image $f_{\mathcal{D}}$, where the edited face belongs to Donald Trump when $T_1$ is set properly.

### 3.2 Threat Model

**Attacker's Ability and Goal**. An increasing number of users are sharing their video content on the Internet through popular social applications such as TikTok and Instagram. However, this

widespread sharing presents a substantial risk to the public's rights, as adversarial users may exploit the capabilities of powerful LDMs to manipulate these videos for illegal purposes. Without loss of generality, we consider two types of malicious video editing. (1) Malicious NSFW editing: the attacker aims to incorporate "not safe for work (NSFW)" elements into a given video, such as blood, corpses, and explicit nudity. (2) Malicious swap editing: the attacker centers around identity substitution within the given video, where one individual's identity (e.g., face) is replaced with that of another to create fake content. These two video editing types are representative for investigation.

**Defender's Ability and Goal**. We assume a challenging black-box scenario, wherein the defender has no information about the *editing models*, *editing methods*, *editing types*, and *editing prompts* that will be used by the malicious user. The defender crafts the perturbations for each frame of the target video based on a public pre-trained LDM, which can be different from the one employed by the attacker. He adds the perturbations to the video and uploads it to the Internet. The goal is to ensure that the perturbed videos can effectively thwart the efforts of malicious editing, e.g., causing the results low-quality, preventing the model from generating NSFW content or identity swapping.

## 4 PRIME

### 4.1 OVERVIEW

Inspired by prior works (Salman et al., 2023; Shan et al., 2023; Le et al., 2023; Rhodes et al., 2023; Zheng et al., 2023), we craft the adversarial perturbation for each frame of the target video to disable its editability by the LDM and break the temporal consistency constraints used in video editing pipelines (Geyer et al., 2024; Yang et al., 2023; Wu et al., 2023) . To overcome the limitations of these solutions, we introduce a series of techniques to achieve three critical properties, as shown in Figure 1. (1) **Zero-shot ability**: the crafted perturbations are effective against different LDM models, editing pipelines, and prompts; (2) **Efficient per-frame perturbing**: we introduce the *fast convergence searching* and the *early stage stopping* mechanisms to significantly reduce the time of perturbing each frame; (3) **Anti dynamic compression**: we design a new scheme to enhance the robustness of the perturbation against the compression in video codecs. Below we give the details of our solution.

### 4.2 ZERO-SHOT ABILITY

With a pre-trained LDM $\mathcal{F}$ consisting of an encoder $\mathcal{E}$, a decoder $\mathcal{D}$ and a U-Net $\mathcal{U}$, the defender aims to generate the adversarial perturbation for each frame, which can cause $\mathcal{F}$ to produce low-quality results. As the model $\mathcal{F}$ and video editing pipeline used by the defender can be totally different from that of the malicious users, the generated perturbation should be highly transferable across different configurations. Furthermore, considering that the malicious users will adopt various prompts to edit the videos, the perturbation should be able to invalidate as many prompts as possible.

To achieve this goal, we follow Photoguard (Salman et al., 2023) to perform a diffusion attack. This indicates that PRIME considers not only $\mathcal{E}$, but also $\mathcal{U}$ and $\mathcal{D}$, and the target is to find a perturbation for each frame, causing the latent features disrupted and $\mathcal{D}(f_{\mathcal{E},0}^s)$ close to a predefined target image. Because the defender has no information about the specific editing pipeline employed by malicious users, we do not consider intricate technologies used in video editing methods, such as cross-frame attention and latent constraints, to make the protection more general. Similar to Photoguard, PRIME depends on the transferability of the perturbation and can be further cooperated with other methods (Kurakin et al., 2017; Athalye et al., 2018; Song et al., 2018) to further improve the effectiveness of the perturbation.

### 4.3 EFFICIENT PER-FRAME PERTURBING

We find that in existing video editing pipelines (Geyer et al., 2024; Yang et al., 2023; Khachatryan et al., 2023; Wu et al., 2023), global attention constraints can rectify artifacts and address imperfections in the structure level of videos. This correction is achieved by referencing information from other frames within the video. Therefore, if not all frames are protected, the generated video will still be maintained at a desirable level. Such observation requires the defender to add the perturbation to every frame. This significantly increases the timing cost as a video can contain a large number

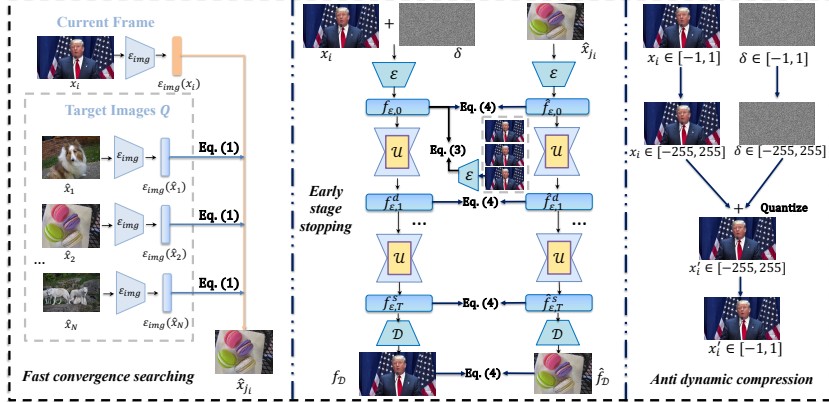

Figure 1: Overview of mechanisms in PRIME. We introduce three new mechanisms to improve effectiveness and efficiency of protecting videos.

of frames. To reduce the time consumption, we introduce two mechanisms, i.e., *fast convergence searching* and *early stage stopping*. The combination of two techniques can reduce more than 90% of timing cost, while maintaining the protection performance.

**Fast convergence searching**. In previous works, the perturbation $\delta$ is oriented to a fixed target image $\hat{x}$. Then, for a protected image $x$, the defender aims to make $\mathcal{F}$ generate a result as close to $\hat{x}$ as possible. The optimization can be described as:

$$\min_\delta d(\hat{x}, \mathcal{F}(x + \delta)),$$

where $d(\cdot, \cdot)$ gives the distance between inputs and $\mathcal{F}(\cdot)$ represents a complete diffusion process to generate images with $\mathcal{F}$. However, we observe that the convergence speed of generating the perturbation differs for different clean images $x$. It will take more optimization steps to obtain the perturbation for $x$ with a slow convergence speed. Therefore, we propose the *fast convergence searching* to find a better target image for each frame. Specifically, given a video $V$, which is constructed using a sequence of frames $V = [x_1, x_2, \ldots, x_n]$, we maintain a queue of potential target images $Q = \{\hat{x}_1, \hat{x}_2, \ldots, \hat{x}_N\}$[1]. For each $x_i$, we select a target image from $Q$ to have the lowest similarity score $s_j$ based on the equation:

$$s_j = \begin{cases} \text{SIM}_{\mathcal{E}_{img}}(x_i, \hat{x}_j), & i = 1 \\ \text{SIM}_{\mathcal{E}_{img}}(x_i, \hat{x}_j) + \text{SIM}_{\mathcal{E}_{img}}(\hat{x}_{j_{i-1}}, \hat{x}_j), & i > 1 \end{cases}, \tag{1}$$

and

$$\text{SIM}_{\mathcal{E}}(x_1, x_2) = \frac{\mathcal{E}(x_1) \otimes \mathcal{E}(x_2)}{|\mathcal{E}(x_1)| \cdot |\mathcal{E}(x_2)|}, j_i = \arg\min_j s_j, \hat{x}_j \in Q, \tag{2}$$

where $\mathcal{E}_{img}$ is the image encoder from the CLIP (Radford et al., 2021), $|\cdot|$ stands for the norm of the vector, $\otimes$ is the matrix multiplication, and $j_i$ is an index for images in $Q$. When $i = 1$, $s_j$ only depends on the first term. Specifically, the second term is to ensure the perturbed continuous frames have different features, to better break the global attention constraints and increase flickers in the edited video. $\hat{x}_{j_{i-1}}$ is the target image for frame $x_{i-1}$. After selection, we obtain $\hat{x}_{j_i}$ for $x_i$. Because the target image and the corresponding frame are very different in the latent space, the convergence speed of the optimization process will be faster at the start.

**Early stage stopping**. We also observe that for a given perturbation budget $\epsilon$ under the $l_p$-norm, increasing the number of optimization iterations will only bring marginal improvement if the perturbation $\delta$ converges in several steps. This inspires us to introduce the *early stage stopping* to further decrease the total time consumption. Specifically, we monitor the similarity $c_k^i$ between the latent generated by $\mathcal{E}$ for the current frame $x_i$ and all previous perturbed frames, i.e.,

$$c_k^i = \max_j \text{SIM}_{\mathcal{E}}(x_i + \delta_k, x_j'), \tag{3}$$

where $\delta_k$ is the perturbation $\delta$ in the $k$-th optimization step, and $x_j'$ is the perturbed frame of $x_j$. When $c_k^i$ does not decrease, we will stop the optimization process and use the perturbation $\delta_k$ for $x_i$.

---

[1]We use the validation set of ImageNet (Deng et al., 2009) as $Q$ in our experiments.

## 4.4 ANTI DYNAMIC COMPRESSION

Different from images, saving videos requires a codec, which applies compression algorithms to balance the file size and video quality. When playing videos on the Internet, it is often more suitable to utilize a variable bitrate as opposed to a constant one. This is due to the fluctuation in the available bandwidth and diverse nature of video content, which demands a dynamic compression ratio applied to each frame to ensure optimal video streaming. Therefore, it is essential to preserve the perturbation information $\delta$ throughout the dynamic compression process to guarantee the effectiveness of the protection. To this end, we introduce a simple solution to achieve anti-dynamic compression. Specifically, we transform the perturbation from the model input space $[-1, 1]$, which is common in previous methods, to the pixel space $[-255, 255]$, and further quantize the values of the perturbation to ensure they have shorter bits. Such operations are the approximated simulation of the compression from the codec. By adding this technique into the protecting process, we can make the perturbation more robust and less sensitive against compression. Through the experiments, we find that our method does not introduce any significant time overhead while increasing the video bitrate by about 8%, which means the video contains more information.

## 4.5 END-TO-END INTEGRATION

The pseudo code of PRIME can be found in Appendix A. We first obtain the most suitable target image $\hat{x}_{j_i}$ for the current frame $x_i$, based on our fast convergence search method. After obtaining $\hat{x}_{j_i}$, we consider calculating the features $\hat{F}$ for it. In Photoguard (Salman et al., 2023), only the final outputs from $\mathcal{D}$ are considered as the features. However, we find that during the diffusion forward process and sampling process, the intermediate results are equally important because we do not have information about the number of diffusion steps $T$ used by the attacker. Disrupting intermediate results can improve the transferability of the perturbation during the diffusion process. On the other hand, calculating the gradient on the intermediate results will not bring additional computing costs, which makes it practical during the protection. To make $\mathcal{U}$ contribute to the forward process, we adopt the DDIM inversion method (Geyer et al., 2024) to predict the noise with $\mathcal{U}$ to replace the original forward process. Therefore, in PRIME, we consider four sources of the features, i.e., $f_{\mathcal{E},0}$ from $\mathcal{E}$, $f_{\mathcal{E},t}^d$ from the DDIM inversion at time step $t$, $f_{\mathcal{E},t}^s$ from the sampling process at time step $t$, and $f_{\mathcal{D}}$ from $\mathcal{D}$. Especially, in the diffusion process, the prompt condition $c_p$ is empty, as we have no information about the editing prompt used by the attackers. Therefore, $\hat{F}$ can be written as

$$\hat{F} = \{\hat{f}_{\mathcal{E},0}, \hat{f}_{\mathcal{E},1}^d, \ldots, \hat{f}_{\mathcal{E},T}^d, \hat{f}_{\mathcal{E},T}^s, \ldots, \hat{f}_{\mathcal{E},1}^s, \hat{f}_{\mathcal{D}}\},$$

which is calculated on the target image $\hat{x}$. When we use $T$ steps in the forward process and sampling process, there will be $T$ features for $f_{\mathcal{E},t}^d$ and $f_{\mathcal{E},t}^s$, respectively. We consider all of them when computing the loss functions.

During the optimization process, we adopt the anti dynamic compression method to add the perturbation $\delta_k$ to the clean frame $x_i$, which is represented by $\prod_\epsilon(x_i + \delta_k)$ under the budget $\epsilon$. Similarly, we compute the features $F$ for the perturbed input $\prod_\epsilon(x_i + \delta_k)$. Then we compute the loss:

$$L = |f_{\mathcal{E},0} - \hat{f}_{\mathcal{E},0}|_1 + |f_{\mathcal{D}} - \hat{f}_{\mathcal{D}}|_1 + \sum_{t=1}^{T}(|f_{\mathcal{E},t}^d - \hat{f}_{\mathcal{E},t}^d|_1 + |f_{\mathcal{E},t}^s - \hat{f}_{\mathcal{E},t}^s|_1), \qquad (4)$$

where $|\cdot|_1$ is the $L_1$-norm. To update the perturbation $\delta_k$, we minimize $L$, i.e., $\min_\delta L$, which gives results closer to the target image $\hat{x}_{j_i}$. After updating $\delta$, we examine the convergence of the optimization process, by computing $c_k^i$ (Eq. (3)). If the optimization process is converged at step $k$, we use $\prod_\epsilon(x_i + \delta_k)$ as $x_i'$ and start to optimize the next frame. Otherwise, we continue to optimize the current frame till the convergence or reaching the attack budget.

Overall, PRIME combines our new proposed mechanisms to accelerate the optimization process and restore more information for the compressed videos. These mechanisms are general and not tailor-made for PRIME, which means future works can directly adopt them to enhance their performance.

| Name | Donald Trump | Drake | Joe Biden | Katy Perry | Messi | Rihanna | Robert Downey Jr. | Ryan Gosling | Scarlett Johansson | Taylor Swift | Sum |
|---|---|---|---|---|---|---|---|---|---|---|---|
| # of clips | 6 | 2 | 4 | 6 | 2 | 2 | 2 | 3 | 2 | 6 | 35 |
| # of total frames | 1027 | 141 | 606 | 574 | 207 | 258 | 285 | 325 | 263 | 430 | 4116 |

Table 1: Details of VIOLENT. It contains 10 famous people and 35 video clips in total.

## 5 EXPERIMENTS

### 5.1 DATA COLLECTION

We notice that currently there are no public standard benchmarks or datasets for the malicious video editing scenarios. This drives us to build a new dataset VIOLENT, i.e., VIdeos fOr maLicious Editing aNd proTection. As summarized in Table 1, VIOLENT consists of videos of 10 celebrities with diverse genders, ages and races. They can be grouped into politicians (i.e., Donald Trump and Joe Biden), singers (i.e., Drake, Katy Perry, Rihanna, and Taylor Swift), actors (i.e., Robert Downey Jr., Ryan Gosling, and Scarlett Johansson), and athletes (i.e., Messi). ***Due to copyright issues, we are unable to make this dataset publicly available.*** To facilitate the reproduction, we provide a list of URLs for these original videos in the supplementary material.

To construct such dataset, we first conduct a manual assessment to identify celebrities who can be perfectly and realistically generated by existing LDMs. We consider diversifying their genders, ages and races. After determining the list of celebrities, we proceed to acquire their videos from various sources on the Internet. Most of these videos are sourced from official channels, while the remaining content is gathered from public channels. All the original videos we collect are in the resolution $1280 \times 720$, which is the most popular format on the Internet. Upon obtaining the videos, we manually edit and cut them into scene-consistent video clips, each comprising tens to hundreds of frames. We carefully filter out clips that contain transitions, illumination changes, or main object changes. Subsequently, we create specific configurations for malicious editing for each clip. For each configuration, we tune the malicious prompts and adjust other hyperparameters used in the video editing pipelines for different LDMs. This entire process requires hundreds of GPU hours of effort.

To create prompts for malicious editing purposes, we follow a very simple and direct template: "[Someone] [Do Something] [Somewhere]". We first create a description for each original video based on this template. For the malicious NSFW editing task, we keep "[Someone]" while change "[Do Something]" and "[Somewhere]". For example, if we want to generate a video in which the person is naked, we replace "[Do Something]" with "is naked" or "is nude". If we want to generate a bloody video, we replace "[Somewhere]" with "in a bloody scene". For the malicious swapping editing task, we only replace "[Someone]" with a new name. For example, if we want to generate a video of Donald Trump based on a video of Joe Biden, we replace "Joe Biden" with "Donald Trump". Furthermore, we adopt the prompt weighting method, Compel com, to manually adjust different weights for "[Someone]", "[Do Something]", and "[Somewhere]" to obtain the best editing results. We design a total of 280 attack configurations, which are used for the subsequent experiments. We also provide a list of such malicious prompts in the supplementary material.

### 5.2 EXPERIMENT SETTINGS

**Editing Models**. We consider public LDMs for high-quality and realistic video generation and editing. After evaluating accessible models on Hugging Face, we manually select four different models, i.e., Stable Diffusion v1-5 (SD1.5), Dreamlike Photoreal 2.0 (DP), HassanBlend1.4 (HB), and RealisticVisionV3.0 (RV). These four representative models have diverse advantages in generating realistic photos under different conditions, such as various illumination and human poses. Our method can be generalized to other models as well.

**Editing Pipelines**. As we consider two types of malicious editing, we choose the most recent and representative open-source editing methods for them, respectively. Specifically, we experimentally find that TokenFlow (Geyer et al., 2024) is suitable for the malicious NSFW editing task, and Rerender A Video (Yang et al., 2023) is suitable for the malicious swap editing task. Our method can be well generalized to other pipelines.

**Protection Implementation**. In our threat model, the defender has no information about the editing models and pipelines used by the malicious users. Therefore, he generates the perturbations based on a public model, and relies on their transferability to prevent malicious editing by other models. We adopt two public models, i.e., Stable Diffusion v1-5 (SD1.5) and Stable Diffusion v2-1 (SD2.1). When crafting the perturbations, we set the maximum number of optimization steps $K$ as 100, and the maximum perturbation size $\epsilon$ as 8 under $l_\infty$-norm for PRIME and Photoguard (Salman et al.,

| Task | SD1.5 | | | | SD2.1 | | | |
|---|---|---|---|---|---|---|---|---|
| | Photoguard | | PRIME | | Photoguard | | PRIME | |
| | **PSNR ↓** | **SSIM ↓** | **PSNR ↓** | **SSIM ↓** | **PSNR ↓** | **SSIM ↓** | **PSNR ↓** | **SSIM ↓** |
| **Malicious NSFW Editing** | **18.30** | 0.64 | 18.32 | **0.57** | 18.58 | 0.65 | 18.42 | **0.57** |
| **Malicious Swapping Editing** | 17.27 | 0.63 | **16.93** | **0.62** | 17.39 | 0.63 | 16.94 | **0.62** |

Table 2: PSNR and SSIM under different protection methods. **Bold** is for the best results, and underline is for the second-best results.

| Task | Original | SD1.5 | | SD2.1 | |
|---|---|---|---|---|---|
| | **VCLIPSim** | Photoguard | PRIME | Photoguard | PRIME |
| | | **VCLIPSim ↓** | | | |
| **Malicious NSFW Editing** | 0.2251 | 0.2151 | **0.2025** | 0.2160 | 0.2028 |
| **Malicious Swap Editing** | 0.2076 | 0.2030 | **0.2022** | 0.2034 | 0.2027 |

Table 3: VCLIPSim under different protection methods. **Bold** is for the best results and underline is for the second-best results.

2023). On the other hand, we notice that Photoguard uses 4 steps in the diffusion sampling process. To maintain the same computing budget, we use $T = 2$ steps in the DDIM inversion and $T = 2$ steps in the sampling process. To form the queue of target images, we use the validation set of ImageNet (Deng et al., 2009) for PRIME. For Photoguard, we use its officially provided target image. The perturbed videos are saved using libx264 codec with variable bitrates and the best quality preference. We resize the resolution of videos into $672 \times 384$. For each video, we only use the first 40 frames to edit.

## 5.3 EVALUATION METRICS

**Subjective Metrics**. We perform a user study and invite 79 persons with different background to assess the generated videos. In our study, we consider six dimensions of human perception, i.e., *Content Consistency*, *Prompt Matching*, *Naturalness*, *Frame Stability*, *Video Quality*, and *Personal Preference*, which are aligned with a recent benchmark, VBench (Huang et al., 2023). Specifically, *Content Consistency* describes the extent to which the edited video maintains the same details as the original video, such as layout, style, movements, and expressions. *Prompt Matching* describes how well the edited video matches the theme of the given prompt, e.g., sex and blood. *Naturalness* describes how plausible the edited video looks. *Frame Stability* measures the temporal consistency of the edited video. *Video Quality* describes the overall quality of the edited video. All these metrics are normalized into the interval of $[1, 5]$. Higher scores mean better performance. For *Personal Preference*, we give survey participants several videos and ask them to pick the one they prefer.

**Objective Metrics**. We consider peak signal-to-noise ratio (PSNR), structural similarity (SSIM), and VCLIPSim. Specifically, PSNR compares the noise ratio in the edited videos w/ and w/o protection. SSIM compares the perceived quality of the edited videos w/ and w/o protection. When computing PSNR and SSIM, we use the original clean videos to generate edited videos as references. VCLIPSim is supported by ViCLIP (Wang et al., 2022), which reflects the similarity between the video and given prompt.

## 5.4 MAIN RESULTS

**Time Consumption and Bitrate Comparison**. We compare the time consumption and bitrate to intuitively validate the effectiveness of our proposed mechanisms in terms of efficiency and robustness against dynamic compression. We record the time cost on one NVIDIA RTX A6000. Notably, both Photoguard and PRIME have a maximum batch size limit of 1 due to GPU memory constraints. For the average time consumption in protecting a video of 40 frames, Photoguard takes 20,500 seconds, while PRIME only takes 1,700 seconds, which is 8.3% of that of Photoguard. For the GPU memory occupation, Photoguard takes about 17GB and PRIME takes about 20GB. The additional occupation in PRIME is the CLIP model used in searching the target images. This indicates that both methods can run on consumer-grade GPUs, such as the RTX 4090. Furthermore, the saved protected videos from Photoguard have an average bitrate of 45021 kbps. These from PRIME have an average bitrate

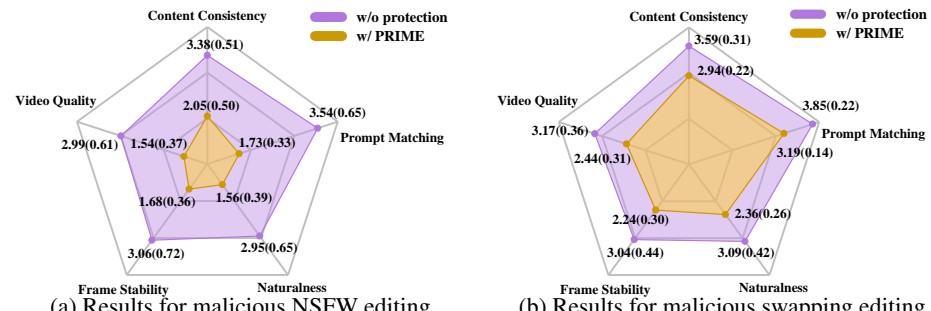

(a) Results for malicious NSFW editing.     (b) Results for malicious swapping editing.

Figure 2: Human evaluation with six subjective metrics on the edited videos of two malicious editing tasks. We show the mean score for each metric and put the standard deviation in brackets.

| Task | Photoguard* | Photoguard | PRIME |
|---|---|---|---|
| **Malicious NSFW Editing** | 35% | 47% | **18%** |
| **Malicious Swap Editing** | 49% | 42% | **9%** |

Table 4: Human preference study. A lower probability is better for the protection method. **Bold** is for the best results. We provide participants with three video clips protected by different methods and ask them to choose the clip that looks the best. Photoguard* stands for the Encoder attack.

of 45776 kbps, which is 8% higher than Photoguard. In summary, the results prove that our proposed PRIME can significantly reduce the time consumption and improve the bitrate, while keeping more information in the saved videos.

**Objective Assessment**. We compute the PSNR and SSIM between the edited videos from the original unprotected ones, and from the original protected ones. In Table 2, we show the protection results on different source models, i.e., SD1.5 and SD2.1, to assess the transferability of the perturbation. The results indicate that even though PRIME costs less time, it achieves better protection performance at the pixel level. Furthermore, PRIME shows better transferability when we compare the results on different editing tasks and source models.

We further compare the prompt consistency of edited videos in Table 3. We observe that PRIME is more effective in breaking the connections between edited frames and the prompt and demonstrates higher transferability between models and pipelines. We notice that although ViCLIP is trained on a large video dataset, it may not be very suitable for evaluating malicious videos. The main reason is that its training set is filtered and contains only a small part of NSFW content and famous people. It will be more reasonable to adopt a fine-tuned ViCLIP as future work. This is beyond the scope of this paper, as we mainly use human evaluations as the primary reference.

**Subjective Assessment**. Since the generated videos mainly aim to mislead humans and cause negative impacts, we perform a user study with the subjective metrics to assess them. We invite multiple individuals to complete a questionnaire. These participates are anonymized for privacy protection. Considering the cost of human evaluation, we provide the quality results for PRIME and human preference results for Photoguard and PRIME. After collecting 140 data samples for each aspect from participants, we present the evaluation results in Figure 2 and Table 4. Based on the quality results in Figure 2, we draw several conclusions. First, existing video editing pipelines can successfully execute the malicious editing tasks and achieve high *Prompt Matching* scores (3.54 out of 5 and 3.85 out of 5) and *Video Quality* scores (2.99 out of 5 and 3.17 out of 5). Second, PRIME can effectively protect videos by decreasing the *Prompt Matching* scores (1.73 out of 5 and 3.19 out of 5) and *Video Quality* scores (1.54 out of 5 and 2.44 out of 5). With our protection, human faces are disrupted and edited frames are less affected by given prompts. Therefore, *Prompt Matching* scores decrease in both tasks. Besides, the stability of the frame and the consistency of the video decrease: there are many flickers between frames, making the video less natural. This is reflected by the *Content Consistency* scores (from 3.38 to 2.05 and from 3.59 to 2.94), *Frame Stability* scores (from 3.06 to 1.68 and from 3.04 to 2.24), and *Naturalness* scores (from 2.95 to 1.56 and from 3.09 to 2.36). Third, for the considered two types of malicious editing, we find that changing the identities of the original videos is easier than adding blood or naked bodies to the original videos, and protecting videos from NSFW editing is easier and more effective. There are mainly two reasons. (1) The swap task is easier because it keeps the layout and other elements unchanged in the videos. The NSFW editing task requires changing the layout and pixel-level details, which is closely related to the capability of

| **Bit Rate** (kbps) | 86 | 623 | 11,533 | 42,881 |
|---|---|---|---|---|
| **PSNR** | 22.18 | 19.02 | 18.79 | 9.71 |
| **SSIM** | 0.79 | 0.78 | 0.63 | 0.27 |

Table 5: The performance under different bit rates.

| **Transformation** | Upscale | JPEG Compression | | | Gaussion Smooth | | |
|---|---|---|---|---|---|---|---|
| | 2x | 30 | 50 | 70 | 1 | 3 | 5 |
| **PSNR** | 26.02 | 24.10 | 22.11 | 20.98 | 26.84 | 27.78 | 30.17 |
| **SSIM** | 0.64 | 0.85 | 0.81 | 0.78 | 0.88 | 0.92 | 0.94 |

Table 6: The performance under different frame transformations.

LDMs to understand the prompt correctly. (2) For the swap task, we adopt additional constraints in the pipeline, such as the optical flow model (Xu et al., 2022) and the canny edge model (Zhang et al., 2023), which enhance the stability and robustness of the edited videos. For NSFW editing, we only use global attention and cross-frame attention to keep the frame consistency because we find that the additional constraints mainly influence the layout and pixel-level details, which conflict with the target of the NSFW editing and make the video editing pipeline give unchanged results for NSFW prompts.

Besides these subjective metrics, we ask the participants to select the video with the best quality among three edited videos, under the protection of Photoguard* (i.e., Encoder attack), Photoguard, and PRIME. The results in Table 4 prove that PRIME will better decrease the quality of the generated videos. More than 80% of the participants think that the generated videos protected by Photoguard have better quality. Therefore, PRIME is more effective in protecting videos from malicious editing. Overall, PRIME achieves better protection performance than Photoguard and reduces the overall time cost. *We leave the visualization results in the supplementary materials, due to the legal considerations*. **Other experiment results can be found in Appendix C–G.** We also discuss the challenges and limitations in Appendix K and L.

### 5.5 ROBUSTNESS STUDY

To study the robustness of our protection method, we consider three variables in video protecting, including bit rates of the videos, video frame transformations, and another video editing method. In Table 5, we first study the bit rates by using different video compression strengths. A lower bit rate corresponds to a higher compression rate. In details, we conducted experiments on a small subset of data using various codec settings with different bit rates. The results indicate that even at very high compression rates, our method remains effective in disrupting video editing techniques, resulting in edited videos of lower quality. These results further demonstrate the robustness of our proposed method against video compression.

In Table 6, we consider three transformations applied directly on each video frame, including super-resolution, JPEG compression, and Gaussian smoothing. Specifically, we evaluate PRIME under 2x image upscaling, JPEG compression with quality settings of 30, 50, and 70, and Gaussian smoothing with scales of 1, 3, and 5. The results indicate that our method demonstrates some robustness against these transformations. However, it is important to note that when the transformations are too strong, such as using very low JPEG compression quality or very high Gaussian smoothing scales, the effectiveness of our protection method is compromised.

Besides DDIM inversion (Song et al., 2021a) based methods used in previous experiments, we consider SDEdit (Meng et al., 2022). The results are as follows: PSNR is 13.34 and SSIM is 0.36. These results indicate that our protection method is effective for SDEdit. Overall, PRIME shows robustness against various factors in video editing pipelines.

### 6 CONCLUSION

In this paper, we investigate the scenario of malicious video editing with generative models, which poses a huge threat to the public. To better protect the videos from malicious editing, we propose a new method, PRIME, which can significantly reduce the timing cost for video protection and improve the bitrate for the protected videos. Based on our evaluation, we prove the advantages of PRIME in protecting videos from editing. We believe that PRIME will promote more future works in protecting videos and portrait rights against illegal usage.

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

---

**Algorithm 1** `PRIME` Algorithm

---

1: **Input:** Video $V = [x_1, x_2, \ldots, x_n]$, target images $Q = \{\hat{x}_1, \hat{x}_2, \ldots, \hat{x}_N\}$, models $\mathcal{E}, \mathcal{D}, \mathcal{U}, \mathcal{E}_{\text{img}}$, diffusion steps $T$, perturbation budget $\epsilon$, optimization steps $K$
2: $V' = []$
3: **for** $i = 1 \to n$ **do**
4:     Obtain $j_i$ based on Eq. (2)
5:     Obtain target feature set $\hat{F}$ for $\hat{x}_{j_i}$
6:     Initialize perturbation $\delta_0$
7:     **for** $k = 1 \to K$ **do**
8:         Obtain feature set $F$ for $\prod_{\epsilon}(x_i + \delta_{k-1})$
9:         Calculate loss based on Eq. (4)
10:        Update $\delta_{k-1} \to \delta_k$
11:        Obtain $c_k^i$ based on Eq. (3)
12:        **if** $c_k^i$ is converged **then**
13:           Append $x_i' = \prod_{\epsilon}(x_i + \delta_k)$ to $V'$
14:           Break
15:        **end if**
16:     **end for**
17: **end for**
18: Return $V'$

---

## A    PSEUDO CODE OF `PRIME`

We give the pseudo code of `PRIME` in Algorithm 1.

## B    HYPERPARAMETERS

We introduce the hyperparameters used in the two video editing pipelines, i.e., Tokenflow and Rerender A Video, as well as the prompt weights. For Tokenflow, the number of DDIM inversion steps is 500, which is its default setting. The guidance scale is 7.5. Other parameters related to the injections are their default values without changing. For Rerender A Video, the interval is set to 1, due to the limited number of frames. The strength of the first frame is 0.8 and the strength of the ControlNet is 0.7. We manually search for these two parameters, which is better than other combinations. The ControlNet we use is the Canny Edge, which we find is better than HED Boundary and Depth. We also set the loose attention, considering the motions in the videos, to obtain a better result. Other hyperparameters are default. We use an additional prompt "RAW photo, subject, (high detailed skin:1.2), 8k uhd, dslr, soft lighting, high quality, film grain, Fujifilm XT3" as a supplement for the prompt.

For both pipelines, the negative prompt is set as "deformed iris, deformed pupils, semi-realistic, cgi, 3d, render, sketch, cartoon, drawing, anime, mutated hands and fingers, deformed, distorted, disfigured, poorly drawn, bad anatomy, wrong anatomy, extra limb, missing limb, floating limbs, disconnected limbs, mutation, mutated, ugly, disgusting, amputation". For Tokenflow, the prompt weights are in the interval of $[1.0, 1.61]$. For Rerender A Video, the prompt weights are in the interval of $[1.0, 1.4]$.

## C    ZERO-SHOT CAPABILITY ON DIFFERENT MODELS

Compared with Photoguard, our method causes the edited videos to have lower quality and more misaligned semantics crossing various unseen models. Besides, the protected videos are unified against both editing types, indicating the generalization of our protection. We have studied the zero-shot capability on different diffusion models. In Table 2 and Table 3 of our main paper, we show the *average* results on four diffusion models (Stable Diffusion v1-5, Dreamlike Photoreal 2.0, HassanBlend1.4, and RealisticVisionV3.0). In Tables 7 and 8, we provide more detailed results on each diffusion model, protected based on SD1.5 and SD2.1, respectively.

Above results prove the zero-shot ability of our method. Compared with Photoguard, our method makes the edited videos have lower quality and more misaligned semantics crossing various unseen

| Model | Original | SD1.5 | | | | | | SD2.1 | | | | | |
|---|---|---|---|---|---|---|---|---|---|---|---|---|---|
| | | Photoguard | | | PRIME | | | Photoguard | | | PRIME | | |
| | VCLIPSim | VCLIPSim | PSNR | SSIM | VCLIPSim | PSNR | SSIM | VCLIPSim | PSNR | SSIM | VCLIPSim | PSNR | SSIM |
| SD1.5 | 0.2275 | 0.2159 | 20.22 | 0.71 | **0.2126** | 20.04 | **0.63** | 0.2211 | 20.69 | 0.72 | 0.2137 | 20.17 | **0.63** |
| HassanBlend1.4 | 0.2274 | 0.2193 | 19.70 | 0.67 | 0.2105 | **19.18** | **0.58** | 0.2246 | 19.79 | 0.68 | **0.2103** | 19.27 | **0.58** |
| RealisticVisionV3.0 | 0.2311 | 0.2199 | **15.02** | 0.56 | 0.1977 | 15.71 | **0.47** | 0.2153 | 15.18 | 0.55 | **0.1966** | 15.75 | **0.47** |
| Dreamlike Photoreal 2.0 | 0.2143 | 0.2052 | **18.26** | 0.62 | **0.1892** | 18.33 | **0.57** | 0.2031 | 18.65 | 0.64 | 0.1907 | 18.48 | **0.57** |

Table 7: Results when protecting videos from Malicious NSFW Editing. **Bold** for the best results and underline for the second-best results.

| Model | Original | SD1.5 | | | | | | SD2.1 | | | | | |
|---|---|---|---|---|---|---|---|---|---|---|---|---|---|
| | | Photoguard | | | PRIME | | | Photoguard | | | PRIME | | |
| | VCLIPSim | VCLIPSim | PSNR | SSIM | VCLIPSim | PSNR | SSIM | VCLIPSim | PSNR | SSIM | VCLIPSim | PSNR | SSIM |
| SD1.5 | 0.2064 | 0.2027 | 18.31 | 0.70 | 0.2040 | 17.83 | **0.68** | 0.2093 | 18.47 | 0.69 | **0.2018** | 17.91 | **0.68** |
| HassanBlend1.4 | 0.2110 | 0.2076 | 18.31 | 0.72 | **0.2047** | 18.06 | **0.70** | 0.2057 | 18.60 | 0.72 | 0.2066 | **18.03** | **0.70** |
| RealisticVisionV3.0 | 0.2102 | 0.2085 | 16.73 | 0.57 | 0.2097 | **16.20** | **0.55** | **0.2071** | 16.70 | 0.56 | 0.2122 | **16.35** | **0.55** |
| Dreamlike Photoreal 2.0 | 0.2030 | 0.1931 | 15.74 | 0.55 | 0.1904 | 15.60 | **0.54** | 0.1917 | 15.78 | 0.55 | **0.1903** | 15.45 | **0.54** |

Table 8: Results when protecting videos from Malicious Swapping Editing. **Bold** for the best results and underline for the second-best results.

models. On the other hand, the protected videos are unified against both editing types, which means our protection is general.

# D  SCALABILITY OF PRIME ON DIFFERENT LENGTHS AND RESOLUTION OF VIDEOS

We conducted several experiments to study the scalability of PRIME on different video resolutions and lengths.

Our method is frame-by-frame. Therefore, for videos with different lengths, the time consumption is linear with the video length. It takes 1,700 seconds, 2,600 seconds, 3,500 seconds, and 4,300 seconds to process a video containing 40 frames, 60 frames, 80 frames, and 100 frames, respectively. The slight difference in the processing time per frame is mainly due to the convergence speed of different frames. On the other hand, increasing the frames will not cost more GPU memory. Therefore, our method is efficient.

When increasing the video resolution, we find the main bottleneck is GPU memory instead of time cost, as well-known fact that generating high-resolution content will cost more GPU memory. We find that it is impossible to protect videos of 1080P or higher resolutions on one consumer-grade GPU for both Photoguard and PRIME. It is even impossible to edit a 1080P-video on one consumer-grade GPU using advanced editing pipelines, due to the heavy computation of cross-frame attention and other intermediate results. Such GPU memory bottlenecks can be addressed through paralleling models on multiple GPUs.

# E  SCALABILITY OF PRIME ON VIDEOS CONTAINING OTHER CONTENT

We evaluate PRIME on protecting more general videos, such as user's content, based on the example videos used by Tokenflow. The results (**PSNR is 18.33**, and **SSIM is 0.73**) prove PRIME can protect these videos from being edited by reducing the video quality. It is because PRIME does not rely on any specific prior assumption of the protected videos, including their content and editing prompts. Therefore, PRIME can protect other content as well.

# F  HUMAN EVALUATION RESULTS ON PHOTOGUARD

Due to the high cost of conducting human assessments, we collect 69 responses for the Photoguard video quality assessment. We provide detailed results besides Figure 2 to better show the advantages of PRIME in Tables 9 and 10. Comparing the results of Photoguard and PRIME, we find that PRIME achieves better protection performance on both malicious editing tasks by heavily decreasing the quality of edited videos.

| Protection | Content Consistency | Prompt Matching | Naturalness | Frame Stability | Video Quality |
|---|---|---|---|---|---|
| No Protection | 3.38(0.51) | 3.54(0.65) | 2.95(0.65) | 3.06(0.72) | 2.99(0.61) |
| Photoguard | 3.19(0.21) | 3.03(0.52) | 2.46(0.27) | 2.59(0.37) | 2.57(0.37) |
| PRIME | **2.05(0.50)** | **1.73(0.33)** | **1.56(0.39)** | **1.68(0.36)** | **1.54(0.37)** |

Table 9: Human evaluation on six subjective metrics for Malicious NSFW Editing. We show the mean score for each metric and put the standard deviation in brackets. The results are calculated on the edited videos.

| Protection | Content Consistency | Prompt Matching | Naturalness | Frame Stability | Video Quality |
|---|---|---|---|---|---|
| No Protection | 3.59(0.31) | 3.85(0.22) | 3.09(0.42) | 3.04(0.44) | 3.17(0.36) |
| Photoguard | 3.42(0.11) | 3.65(0.13) | 2.84(0.02) | 2.86(0.16) | 2.90(0.07) |
| PRIME | **2.94(0.22)** | **3.19(0.14)** | **2.36(0.26)** | **2.24(0.30)** | **2.44(0.31)** |

Table 10: Human evaluation on six subjective metrics for Malicious Swapping Editing. We show the mean score for each metric and put the standard deviation in brackets. The results are calculated on the edited videos.

## G    HUMAN EVALUATION RESULTS ON VIDEO AUTHENTICITY

Besides Table 4 of human preferences, we collect 84 responses about whether the video is fake (AI generated) or not. In Table 11, we show the collected results. Clearly, most volunteers believe edited videos under the protection of PRIME are generated by AI, due to lots of artifacts in the edited videos.

## H    SCALE OF VIOLENT

In our experiment, the total number of video clips is 35. We implement the video editing task with four diffusion models (Stable Diffusion v1-5, Dreamlike Photoreal 2.0, HassanBlend1.4, and RealisticVisionV3.0) and two editing pipelines (TokenFlow and Rerender A Video). Therefore, the total edited videos are 35 * 4 * 2=280. The main bottleneck of collecting video clips is the huge cost of manually fine-tuning the editing prompts from each clip under different diffusion models and editing pipelines, which has costed us hundreds of GPU hours. Nevertheless, we believe our video clips are sufficient in proving the effectiveness of our solution. Furthermore, in our user study, we select 10 identities with varied races, genders, and other personal information to guarantee the editing results are unbiased.

## I    VISUALIZATION RESULTS

Sorry for the inconvenience that we cannot provide visualization results in the main paper, due to the reasonable concern that the edited content could be leaked to the Internet and cause unnecessary harm to these innocent people. **Therefore, we provide the visualization results in the supplementary materials as a video file.** To further promise that the edited content does not cause further negative impacts, we add visible watermarks and limit the visualization similarity of the faces.

## J    ACCESSIBLE RESOURCES FOR VIOLENT

Although it could be illegal to release the whole dataset, we consider only providing the configuration files used in our experiments. These configurations are packed in our supplementary materials. For each model we use in the experiments, we tune the prompts to achieve better editing results. Even though we cannot provide the source videos to the public, protecting these people's rights, these configurations could be useful for future studies. We believe that the readers can obtain enough information about the source videos from these configurations because we describe each video clip we use. On the other hand, for other researchers, who are going to study in this area, we are willing to guide how to construct the dataset after signing a commitment. As we collect these celebrity videos from the Internet, such as YouTube Channels, we believe that other researchers could find videos with similar content as we used. Therefore, it is easy for others to reproduce the same experiment results, under legal consideration. Note that in whatever case, we will not leak the source video clips and re-distribute them.

| Task | Photoguard* | Photoguard | PRIME |
|---|---|---|---|
| **Malicious NSFW Editing** | 52% | 35% | **14%** |
| **Malicious Swap Editing** | 59% | 28% | **12%** |

Table 11: Human evaluation on video authenticity. The normalized results of answers that the given video is not generated by AI. A lower value means more people believe these videos are generated by AI. **Bold** for the best results. Photoguard* stands for the Encoder attack.

## K    CHALLENGES IN HUMAN EVALUATION

We follow the previous methods in Tokenflow and Rerender A Video to evaluate the edited videos, i.e., using both objective and subjective metrices. To quantify subjective measures, we invited volunteers to fill out a questionnaire. For each metric, we provide a valid interval from 1 to 5. Before the evaluation, we first give the volunteers an unedited video correlating to the celebrity, which will be a standard for their evaluation. Then, the evaluation of edited videos is based on the preferences of volunteers. And we make the best effort to avoid biases in the human evaluation results. Specifically, we strictly select volunteers with varied genders, ages, and culture backgrounds. Furthermore, we include the standard deviation for each metric in the human evaluation results to clearly present the diversity of the results. With these measures, the evaluations will be more general to all the people. Currently, using automated evaluation is still an open problem in content generation area, which is out of the scope of our paper.

## L    LIMITATION

The main limitations of PRIME are from two aspects. First, we consider a black-box protection scenario without any information about the editing methods. It will decrease the performance of PRIME for some more advanced editing methods, such as OpenAI's Sora[2]. Second, although PRIME only takes 8% of the running time of Photoguard, it still cannot protect a livestreaming video, which could be a very important research direction in the future.

Additionally, we notice that Sora from OpenAI adopts DiT (Peebles & Xie, 2023), which would patchify the given frames to patches. This could be a method to circumvent the protection. However, as Sora is not released to the public, we cannot verify this point. But we think it is possible to protect videos against DiTs, just like previous works (Aldahdooh et al., 2021; Naseer et al., 2021) attacking ViTs.

## M    SOCIAL IMPACTS, ETHICAL AND LEGAL CONCERNS

It has been a very long story to protect portrait rights and other intellectual property since the generative models became popular on the Internet. For example, a very recent case is the AI-generated Taylor Swift photos circulating on social networks, which raises the attention of many popular presses, such as BBC, and even the White House. Not just for famous people, advanced generative models can affect normal people's photos as well, due to editing technologies and personalization technologies. Such powerful generative models can be used for illegal use. Therefore, adding watermarks into the generative models to detect AI-generated content, adding perturbation to images to immunize editing, and detecting illegal content before returning the results to users are three main solutions in addressing the ethical and legal concerns of using generative AIs.

Similar to generating images or editing images, generating videos and editing videos become possible with advanced generative AI models. Although it is still at the beginning phase, we find that the rising video editing pipelines equipped with advanced latent diffusion models are able to do some operations, such as swapping faces, adding naked bodies, and replacing backgrounds. This finding motivates us to study the risks for the public under the threat of such video editing operations.

First, at the forefront of these concerns is collecting other people's selfie videos is easy. Sharing selfie videos to social applications, such as TikTok and Instagram, is very popular among Gen Z (people who are born between the mid-to-late 1990s and the early 2010s). People's desire to share

---
[2]https://openai.com/index/sora

becomes an exact threat to themselves. Because others can easily download their videos from social applications and edit them. Based on this point, social applications should protect their users' shared images and videos.

The second concern is from an ethical standpoint for these AI-generated videos and AI-edited videos, which pose a significant threat to the veracity of information and encourage the dissemination of misinformation, defamation, and deleterious content. These influences jeopardize the integrity of information dissemination, eroding the public's trust in online platforms, such as the media and social applications, as reliable sources of information. Furthermore, malicious editing may contravene privacy statutes, transgress intellectual property safeguards, and incur charges of defamation. This complex legal terrain demands a nuanced examination of liability, accountability, and the potential gaps in existing legal frameworks that may be exploited by malicious actors. Especially, the existing legal system has not provided enough evidence and support for the judges to determine a potential crime.

The third concern is that platforms that host user-generated content can promote and encourage users to create these AI-generated videos and AI-edited videos. These platforms should provide users rights to generate and edit videos and supervise users' activities to avoid providing potentially illegal or harmful videos. We find that some existing platforms, such as Gen-1 (Run), provide a detection model to detect whether the generated video contains illegal or harmful content. However, other platforms, such as Pika (Pik)[3], do not detect the outputs, making it possible to generate videos containing naked bodies and blood.

In addition, the issues caused by the illegal and harmful videos generated by AI encompass broader societal implications due to the Internet. For example, the unchecked proliferation of illegal and harmful videos can sow discord, amplify existing divisions, and potentially incite harm between people. It is possible to create AI-generated videos, such as an imitation of the Murder of George Floyd, causing arguments on the Internet between different races.

Due to the flaws and imperfections in the current legal system and social framework, we call on users to protect their videos and legitimate rights and interests. We propose a protection method to help users who are willing to share their selfie videos on social media to protect these videos. With our protection, it will be easier for humans to recognize AI-edited videos. In this way, we hope that our approach can provide an interim solution until a full legal and regulatory system is proposed.

---

[3]We tested the Beta test version.

