# OpenReview forum: "PRIME: Protect Your Videos From Malicious Editing"
_ICLR.cc/2025/Conference — ICLR 2025 Conference Withdrawn Submission_

### Official Review · Reviewer_v93Z · 2024-10-31

**Soundness:** 2
**Presentation:** 3
**Contribution:** 2
**Rating:** 5
**Confidence:** 4

**Summary:**

The authors propose a protection method for video editing, namely PRIME, which can reduce the timing cost for video protection and improve the bitrate for the protected videos. A user study is conducted.

**Strengths:**

- The paper is easy to understand
- The experiments demonstrate the effectivenesss

**Weaknesses:**

- Technical novelty is limited. There is no detailed explanation of the difference between the proposed methods and the existing ones.
- Ablation studies are missing. There is no analysis of the strengths claimed in the method sections. For example, no computational time analysis on "EFFICIENT PER-FRAME PERTURBING".
- Explanation is needed for robustness study. I am not sure if the number in the table stands for the comparison between the original frame and the protected frame or the original frame and the edited frame.
- The visualization is not satisfactory. The perturbation in the protected video is perceptible and the edited video is not totally destroyed.
- Comparison methods are lacking. Only PhotoGuard is compared.

**Questions:**

See the Weakness.

---

### Official Review · Reviewer_RHVP · 2024-11-03

**Soundness:** 2
**Presentation:** 2
**Contribution:** 1
**Rating:** 3
**Confidence:** 4

**Summary:**

This paper presents a method to protect videos from malicious editing by injecting perturbations into video frames. It follows the approach in Photoguard, and proposes two techniques, fast convergence searching and early stage stopping, to improve the efficiency for videos. It evaluates the proposed method on a custom dataset of celebrity videos and shows improved efficiency and protection compared to prior work.

**Strengths:**

This paper is addressing an important problem given that video editing technologies are advancing rapidly.
This paper is generally clear in its presentation.

**Weaknesses:**

Limited technical novelty and trivial solution:
- The core method is largely an adaptation of Photoguard to videos.
- The "fast convergence searching" is essentially just target image selection optimization. And it is not clear how to select the target image pool, and how different target image pools will influence the performance. In addition, this paper should compare with non-targeted optimization, i.e. add perturbation to maximize the distance between original frame and perturbed frame.
- Early stopping is a standard ML technique with no significant innovation.
- The anti-compression technique is a straightforward quantization approach.

Experimental concerns:
- The evaluation dataset is small (35 clips) and not publicly available. It only includes celebrity videos, which does not represent the broad distribution of videos in practice.
- It does not show protection against video editing techniques besides diffusion model based tools. There is no evidence that this method is still effective for more advanced and proprietary video editing models, like pika, runway gen-3, and sora.

Performance is not impressive:
- There are no qualitative results in paper but from the videos, we can see noticeable noise in the perturbed videos  in the supplementary. The authors should include qualitative results in the main paper for easier visual assessment.
- Even for clean videos, the edited effect looks terrible. Making already bad performing video editing models perform worse is not impressive.

**Questions:**

1. How would the method perform against more advanced video editing architectures not tested in the paper?

1. Current perturbations are noticeable. What is the minimum perturbation required for effective protection? Is there a theoretical bound?

1. Can you provide more details about the human evaluation methodology? How were participants selected and potential biases controlled?

---

### Official Review · Reviewer_ewRN · 2024-11-04

**Soundness:** 3
**Presentation:** 3
**Contribution:** 3
**Rating:** 6
**Confidence:** 4

**Summary:**

This paper proposes a framework named PRIME that can inject adversarial noise into video frames, so that the video cannot be correctly edited by some video editing pipelines like TokenFlow and Render-A-Video. The authors propose some tricks to make the attack stronger and more effective, e.g. fast convergence searching and early stopping. The paper is well-written and easy to read, both the numeric results and visualizations in the supp material show effectiveness of the proposed algorithm.

**Strengths:**

- The paper is well-written and easy to understand.
- It focuses on a critical problem, large generative models can be maliciously used to edit our videos without permission, we should act first and propose some protection before publishing our videos on the internet.
- The proposed method is effective according to the experimental results. Some novel attacking tricks are included e.g. early stopping, fast searching and anti-compression.

**Weaknesses:**

## Clarifications
- The black-box setting is not that convincing, because as far as i know, TokenFlow and Render-A-Video also use stable diffusion backbones (same visual encoder as the attacking pipelines).


## Methods
- Eq (4) seems to be slow, can we just sample $t$ instead of summing up all the $T$.

## Experiments
- Photoguard can be a baseline, must there are many stronger protection baselines than PhotoGuard e.g. Mist [1] and SDS [2].
- No ablation studies are conducted to show that the proposed fast convergence and early stopping really matter.

## Others


The author should introduce more about the targeted editing pipeline like TokenFlow, since it is different from Image-to-Image editing pipelines.


[1] Mist: Towards improved adversarial examples for diffusion models

[2] Toward effective protection against diffusion-based mimicry through score distillation

**Questions:**

No

---

### Official Review · Reviewer_2R3n · 2024-11-05

**Soundness:** 3
**Presentation:** 3
**Contribution:** 2
**Rating:** 3
**Confidence:** 4

**Summary:**

The paper focuses on creating adversarial perturbations for private videos to prevent malicious editing by open-source diffusion models.
The paper argues that existing image protection methods are ineffective for video protection due to their high computational costs and limited robustness.
To address this, the paper proposes a video protection framework named PRIME, which employs fast convergence searching and early stopping for efficiency, and an anti-dynamic compression strategy for robustness.
Experimental results indicate that PRIME provides more robust video protection with significantly lower computational costs compared to image-based protections like PhotoGuard.

**Strengths:**

1. The issue of protecting videos from malicious editing is of high significance. The paper highlights the limitations of applying image protection methods to videos and underscores the need for dedicated video protection strategies.
2. The experiments demonstrate that PRIME is better suited for video protection than image-based methods (PhotoGuard), offering lower computational costs and enhanced robustness.

**Weaknesses:**

1. The paper lacks ablation studies. PRIME comprises various optimization strategies: attacking the entire diffusion process, employing fast convergence searching with early stopping, and utilizing anti-dynamic compression. However, the paper does not provide ablation studies to verify the necessity and effectiveness of these strategies. Given the marginal performance improvement of PRIME over PhotoGuard, as shown in Tables 2 and 3, conducting ablation studies is crucial to understand which strategies significantly enhance performance.
2. The paper provides insufficient details for some experiments. In Tables 5 and 6, the choice to experiment only on a subset of datasets, rather than all available datasets, is not well-explained. This issue is also apparent in Appendix Section E, where the authors casually mention evaluating PRIME on "more general videos" without specifying the models, datasets, or hyper-parameter settings used. A more detailed explanation of these choices would strengthen the paper.
3. It is advisable to include video-text alignment (VCLIPSim) as an evaluation metric in Tables 5 and 6. While poor generation quality does not necessarily indicate the absence of undesirable elements like NSFW content, comparing the performance of PRIME with PhotoGuard and ‘w/o protection’ could provide deeper insights into its efficacy.
4. The paper only compares PRIME to PhotoGuard, but other image protection methods like Anti-Dreambooth, AdvDM, and Mist have shown greater effectiveness in certain scenarios. It would be beneficial to explore whether PRIME could be integrated with these methods and if it offers any additional advantages over them.

- [1] Le et al. Anti-dreambooth: Protecting users from personalized text-to-image synthesis.
- [2] Liang et al. Adversarial Example Does Good: Preventing Painting Imitation from Diffusion Models via Adversarial Examples.
- [3] Liang et al. Mist: Towards Improved Adversarial Examples for Diffusion Models.

**Questions:**

See in Weakness

---

### Note · Authors · 2024-11-14

I have read and agree with the venue's withdrawal policy on behalf of myself and my co-authors.